# Spiritual Care in Palliative Care: A Physician's Perspective

## Marie-José H.E. Gijsberts

End-of-Life Research Group, Vrije Universiteit Brussel & Ghent University, 1090 Brussels, Belgium; marie-jose.gijsberts@vub.be

**Abstract:** Palliative care is defined as 'an approach that improves the quality of life of patients and their families who are facing problems associated with life-threatening illness. It prevents and relieves suffering through identification, assessment and treatment of pain and other problems, whether physical, psychosocial or spiritual'. As a palliative care physician, I aim to explore and meet the needs of my patients and their loved ones. As I am specifically trained as a specialist in assessing and treating 'pain and other physical symptoms', in psychological, social and spiritual issues, I am a generalist. Two approaches are described to assess spiritual needs in palliative care: The first is adjoining the analytic concept of the four dimensions of palliative care: using an instrument, measuring spiritual well-being or spiritual needs, and measuring the quality of life, with specific attention to spiritual issues. Second, a holistic approach is promoted, with openness to the patients' narrative of their life, disease and suffering. In the integrity of the clinical encounter, medical, ethical and spiritual issues may be discussed. Broadening our clinical language with ethical, psychosocial, and spiritual vocabulary is mandatory, and self-reflection, interdisciplinary collaboration and specific interdisciplinary training may be supportive to develop such a clinical language.

**Keywords:** palliative care; end of life; spirituality; spiritual care; meaning; communication; assessment

## 1. Introduction

The World Health Organization uses a comprehensive definition of palliative care: 'an approach that improves the quality of life of patients (adults and children) and their families who are facing problems associated with life-threatening illness. It prevents and relieves suffering through the early identification, correct assessment and treatment of pain and other problems, whether physical, psychosocial or spiritual.' (WHO Definition Palliative Care 2022 available online: https://www.who.int/news-room/fact-sheets/detail/palliative-care, accessed on 22 February 2022).

As a palliative care physician, I aim to explore and meet the needs of my patients and their loved ones as carefully as possible. This may be a challenging undertaking, as I am specifically trained as a specialist in assessing and treating 'pain and other physical symptoms'. In psychosocial and spiritual issues, I consider myself to be a generalist, with psychologists and social workers the specialists in psycho-social care and health care chaplains the specialists on care for spiritual needs.

In this paper, I would like to share my thoughts on possible approaches concerning identifying spiritual needs and providing spiritual care in palliative care by generalists like me and how interdisciplinary collaboration and specific training may be beneficial, first and foremost to our patients.

## 2. How Spiritual Issues and Spiritual Distress May Be Presented to a Physician

As a palliative care specialist, I also work as a consultant for physicians and nurses who are confronted with complex palliative problems. In a recent consultation, a general practitioner called me for advice on one of her palliative patients. The consultation gives an example of how spiritual issues may be part of a clinical encounter between physician and patient [Box 1].

**Box 1.** Example of spiritual issues as part of a clinical encounter between physician and patient.

> The consultation concerned a 34-year-old woman. She had had a screening for cervical cancer a year earlier, just after her second pregnancy, and the test result could not have been worse: Pap 5, stage IV. She had metastasis in her lungs and abdomen/peritoneum, which led to ascites. Until recently, she was still under the treatment of the oncologist, and she received palliative chemotherapy and an experimental treatment. The oncologist had stopped the treatment because it was not effective anymore: there was progression in tumour growth, and the metastasis in her peritoneum produced increasingly more ascites, up to 1.5 L per day. She had a drain, and she was able to tap the fluid herself. Things were going reasonably well; she had just returned from a camping holiday with her family. However, since yesterday, she was increasingly uncomfortable due to nausea and vomiting, especially in the night. She had been vomiting all night. She was exhausted. The GP describes her as a strong woman, and there are no signs of anxiety or depression. She has indicated several times that she wants to be with her husband and her two young children for as long as possible.

Of course, there may be patients who directly indicate to their physician that they want to discuss their spiritual needs (Gijsberts et al. 2020), but mostly, the needs of the patient are indicated to the physician by the patient in a similar way as described in the consultation: somewhere in the patient's story, spiritual needs may be expressed.

### 3. Spiritual Care: One of the Four Dimensions of Palliative Care

In the Dutch Guideline Spiritual Care in Palliative Care, a model is shown (Figure 1) that represents the relationship between the four dimensions of palliative care (Dutch Guideline Spiritual Care in Palliative Care 2018, available online: https://www.pallialine.nl/uploaded/docs/Zingeving/Existential_and_spiritual_aspects_of_palliative_care_zonder_linken.pdf?u=1SxZX1, accessed on 22 February 2022). The spiritual dimension is the least visible and the most intimate dimension in this model. In addition, the red and blue arrows indicate a dynamic interaction between the four dimensions of palliative care. These dimensions, which are also explicitly described in the WHO definition, may be distinguished but cannot be separated. Recent studies show that this may even be difficult, especially to distinguish the psycho-social and spiritual dimensions (Gijsberts et al. 2020; Lormans et al. 2021).

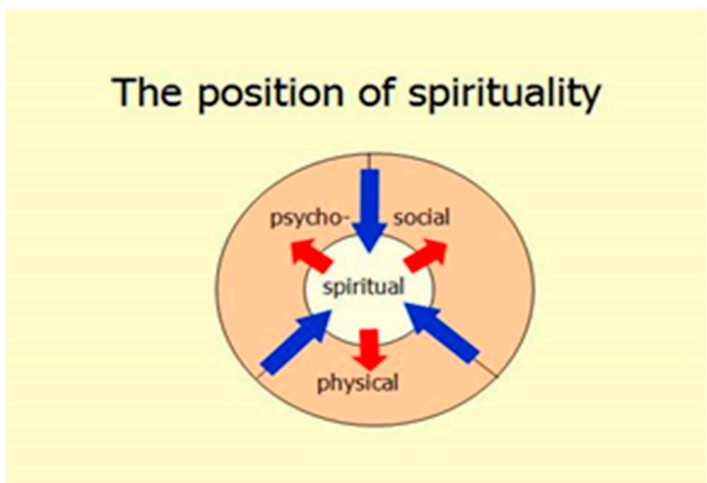

**Figure 1.** The relation between the four dimensions of palliative care.

One of the ways to approach a patient's spiritual needs is by trying to identify them, for instance, by using instruments that are validated to measure spiritual needs or spiritual well-being in palliative care (Gijsberts et al. 2011). Some instruments measure the quality of life at the end of life and include items on the spiritual dimension, such as the Quality of Life at the End of Life (QUAL-E) measure (Steinhauser et al. 2004) and the Missoula Vitas QOL (MVQOLI; Byock and Merriman 1998); other instruments focus specifically on

spiritual issues, such as the Spiritual Needs Inventory (SNI; Hermann 2006) and the JAREL Spiritual Well-being Scale (Hungelmann et al. 1996).

If distinguishing psycho-social needs from spiritual needs is complex, why do we still use models in palliative care, representing the four dimensions, as though we are divisible into 'parts'? These dimensions are four different ways to talk and think about the suffering of our patient. This may be helpful, for instance, to determine the best professional support for the patient by discussing and analyzing the patient's illness, symptoms and treatment options in this way: e.g., if a patient has a low score on QUAL-E items that refer to spiritual well-being or a low score on the SNI, a referral to a health care chaplain or a specialized psychologist or social worker should be considered.

For example, in the patient described in the consultation [Box 1], the suffering from the (physical) disease and the symptoms are obvious: the discomfort from the nausea, the vomiting, the ascites and the exhaustion. In addition, it is palpable that these symptoms interact with the way she can say goodbye to her husband and children and to come to terms with this situation, including the almost unthinkable idea that she will have to leave her children behind. These needs reflect challenges that may be 'labelled' as psycho-social, as well as existential, as spiritual needs. They reminded me of the EAPC definition of spirituality, which comprises 'Existential challenges (e.g. questions concerning identity, meaning, suffering and death, guilt and shame, reconciliation and forgiveness, freedom and responsibility, hope and despair, love and joy)' (EAPC Definition of Spiritual Care 2011: https://www.eapcnet.eu/eapc-groups/reference/spiritual-care/, accessed on 22 February 2022). Medical treatment of the nausea and vomiting may relieve suffering and make it easier for her to focus on the process of letting go and saying goodbye to her loved ones. This is an example of how a theoretical model of the four dimensions of palliative care may be helpful in relieving the suffering of patients.

## 4. Palliative Care as Holistic Care, Welcoming the Patient's Narrative and the Concept of Integrity

There is another, more in-depth and sometimes more time-consuming, approach to assess whether the patient's suffering includes spiritual suffering, which aligns with palliative care being holistic care. For me, this approach implies that when I have a consultation with a patient, I aim to encounter them as a 'whole person' (not as a conglomerate of four dimensions). I try to fully understand the patient's suffering, and therefore, I have to enable my patients to share the narrative of their situation with me as they know and experience it. My role in this clinical encounter will be to be fully aware and open, as the patient is only really knowable through sharing this narrative. I may invite and encourage the patient, using questions that are expansively open ended and attuned to the patient. For instance, in the patient described in the consultation, questions such as 'Tell me about your vomiting' and 'Tell me what this is like for you' could be appropriate (Mohrmann and Shepherd 2011).

I remember, as an intern, we were taught to ask these sorts of questions, and I was aware of the intimate territory these questions referred to, and the depth of this aspect of being human initially made me feel a little insecure and uncomfortable, as I felt I had to open myself to this depth inside of me. Luckily, I was part of an interesting group of interns. Amongst ourselves, we would playfully and safely experiment with these questions and disclose and explore this territory in ourselves and in each other. For instance, when one of us was looking exhausted on Monday morning and said, 'Ooh, I'm so tired', we would ask, 'Tell us more about it.' This started out as a joke, but after a year, we became increasingly familiar with welcoming each other's stories, with listening to the narrative of each other's lives. In addition, we learned, in the conversations with our patients, that asking these open-ended questions and being a little patient may make an enormous difference in helping the patients to reveal as much of the whole of their narrative as possible and letting them know that they have been heard, instead of taking a shortcut and moving the conversation forward a little faster, with data from questionnaires and categorizing questions. The latter may even leave patients with a feeling that they were not able to

truly represent their story and that a treatment that was discussed and recommended was possibly based on this false representation. So, there is a big difference between taking a medical history and hearing the patient's story; it is the difference between the disease, the symptoms and signs, and the patient's narrative, which we should honour in the way it is shared with us, whether coherent or not, whether confusing, inconsistent or disturbing or not (Waitzkin 1993; Frank 1995).

For me, the 'wholeness' of the patient's narrative also relates to the concept of integrity of the clinical encounter. Integrity is defined as (1) 'the quality of being whole and complete' as well as (2) 'the quality of being honest and having principles' (The Cambridge Dictionary Definition of Integrity 2022: https://dictionary.cambridge.org/dictionary/english/integrity, accessed on 22 February 2022). This integrity of the clinical encounter means that my conversation with the patient will comprise medicinal, ethical and spiritual issues. The patient's narrative, and our invitation to share more on issues that are close to their heart, should move seamlessly from one topic to another, and from this naturally flowing conversation, the needs that are the most important to the patient, including spiritual needs and issues, will emerge in this way. Most clinicians will recognize that an integer consultation with a patient will engage medical knowledge and information, the patient's spiritual understanding and valuing of the current situation, and ethical choices and deliberations; one always entails the others because they are not separable (Mohrmann 2015). For instance, in the young woman presented in the consultation [Box 1], the conversation can and must move smoothly, intelligibly and simultaneously among MRI results of her abdomen; the significance of a potentially shortened lifespan; the tapping of the ascites fluid; the meaning of losing her children; the obligations she feels to self and family; decisions about medication that would make her possibly more drowsy; and sources of support, such as her husband and her sister—medical information, spiritual valuing, and ethical aspects and difficult choices, all converging in this one clinical encounter.

## 5. Developing a Clinical Language and the Importance of Interdisciplinary Collaboration and Training

How do we, as physicians, develop a clinical language that moves in a natural way between all the different questions and needs of the patient, being a specialist in one aspect, the medical in my case, but a generalist in other aspects and dimensions? For most of us, it will start as an 'internal' pidgin in which we have our foundations in own professional (and personal) vocabulary, and take elements of languages from other disciplines, such as ethics, social work, psychology and spirituality, and step by step develop a language that comprises increasingly more aspects. My clinical language is rooted in the vocabulary of geriatric and palliative medicine, in explaining and discussing symptoms and options for symptom management, as well as ethical considerations. In the process of enriching and expanding our clinical language, colleagues from other professional disciplines, including health care chaplains, may be inspiring. For instance, social workers have made the concept of 'total pain' palpable for me, e.g., by showing how financial problems and relational challenges may contribute to the way the patient experiences pain or nausea, and the chaplain introduced the importance of paying attention to the patient's use of sayings and proverbs and explore their meaning, such as 'When I heard the diagnosis, my heart stopped', recognizing anticipating grief and, of course, welcoming the patient's narrative.

The WHO definition of palliative care promotes a team approach to supporting patients and their caregivers. When we have an opportunity to collaborate with other professional disciplines, and when your colleagues from other professional disciplines welcome a patient with a similar openness, this provides an opportunity in an interdisciplinary meeting to get an even more full and integrated understanding of the patient. When this understanding shows that a patient has spiritual needs, a referral to a specialist in spiritual care and support from a health care chaplain or a specialized psychologist or social worker should be considered. However, in addition, colleagues in such a team meeting may inspire each other to enrich your own clinical language. As a physician, I consider myself to

be a generalist concerning spiritual issues, with my predominantly somatic background. Inspiration by a health care chaplain in the team may be key in developing our own clinical language, with more understanding and skills to 'weave' specific attention for spiritual issues into our understanding of the patient. As spiritual and psycho-social needs are difficult to distinguish, we see a promising and understandable development: In recent years, psychologists, psychiatrists and medical socials workers have also developed interventions to support/treat patients, such as Meaning Centred Therapy (Breitbart et al. 2010), CALM (Rodin et al. 2018) and Dignity Therapy (Chochinov et al. 2011), inspired by the pioneering work of one of the first representatives of the humanistic psychologies, Victor Frankl (Man's Search for Meaning; Frankl 2019), and aiming to support patients in spiritual aspects, such as purpose and meaning in life, and sharing their legacy. The Dutch guideline empowers collaboration between the chaplains and the psycho-social professionals on these needs of patients (Dutch Guideline Spiritual Care in Palliative Care 2018), and it will be interesting to follow up the best practices in how these disciplines succeed in collaboration to serve patients' deepest needs.

In addition, international curricula are developed, such as the 3-day course 'Interprofessional Train-the-Trainer Spiritual Care Education Curriculum' at the George Washington Institute for Spirituality and Health. These training programmes are developed to introduce and promote interdisciplinary collaboration in palliative care for patients with all professional disciplines, to meet their needs, including spiritual needs (Clinician Spiritual Care Education Curriculum (ISPEC) 2018: https://smhs.gwu.edu/spirituality-health/program/transforming-practice-health-settings/interprofessional-spiritual-care-education-curriculum, accessed on 22 February 2022). The training balances knowledge on spirituality and spiritual care, as well as working on the development of an open and listening attitude and practicing skills to communicate about spiritual topics.

## 6. Conclusions

Analytic and holistic approaches are used in palliative care to assess the patient's suffering, including spiritual suffering. A holistic approach implies the challenge in developing a personal clinical language that will enable us to welcome and encourage patients to tell their personal narrative as they know it and to be aware of emerging issues and needs in patients, including spiritual needs. Self-reflection, high-level interdisciplinary collaboration and specific training in interdisciplinary collaboration may be supportive to our journeys to develop this clinical language, which is the bearer of our witnessing the patient's suffering and discussing the support that matches the patient's needs as much as possible.

**Funding:** This research received no external funding.

**Institutional Review Board Statement:** Not applicable.

**Informed Consent Statement:** Not applicable.

**Data Availability Statement:** Not applicable.

**Conflicts of Interest:** The author declares no conflict of interest.

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
