# Peer review of "Spiritual Care in Palliative Care: A Physician’s Perspective"

_religions, doi:10.3390/rel13040323_

Round 1

Reviewer 1 Report

This is an interesting and worthwhile “communication.”

It is not the report of a rigorous empirical study.

It is the write-up of some experience.

The content is quite familiar to this reviewer, an ordained minster with experience in hospital chaplaincy in two cities. (I also have a PhD in Sociology and 35 years of experience in part-time university teaching.) Any reader of clinical pastoral education or pastoral theology journals will recognize that the author is at an early point in the intellectual consideration of interdisciplinary cooperation in health care. There are places in the world where the hoped-for “interdisciplinary collaboration” (line 16) and “clinical language” (line 17) already exist.

However, it is a unique perspective, well reported, and deserves an audience and feedback. I hope it will be read and responded to.

It is quite well written with a few weak spots:

“3. Spiritual Care…” line 49 --- poor syntax

Line 185

“…needs as well as possible.”

Author Response

REVIEWER This is an interesting and worthwhile “communication.”

AUTHOR: Thank you

REVIEWER: It is not the report of a rigorous empirical study. It is the write-up of some experience.

AUTHOR: This is correct, I aim to describe my experience in becoming a ‘generalist in spiritual care’

REVIEWER: The content is quite familiar to this reviewer, an ordained minister with experience in hospital chaplaincy in two cities. (I also have a PhD in Sociology and 35 years of experience in part-time university teaching.) Any reader of clinical pastoral education or pastoral theology journals will recognize that the author is at an early point in the intellectual consideration of interdisciplinary cooperation in health care. There are places in the world where the hoped-for “interdisciplinary collaboration” (line 16) and “clinical language” (line 17) already exist.

AUTHOR: Thank you so much for this feedback.

Of course, in my country there are also multi-disciplinary teams and some of them show  interdisciplinary cooperation that come close to the experience  described by the reviewer, that I would consider to be ‘best practices’ in this field.

Although the reviewer agrees that my contribution is sharing a personal experience and development as a palliative care physician to promote this collaboration and developing a clinical language, I considered to add to the conclusions:

Pg 5 line 185: A study in best practices of interdisciplinary collaboration, including facilitating and hindering factors, may also support and inspire this development.

Would the reviewer consider this as an addition that also does justice to these best practices and may put this paper in a broader perspective?

REVIEWER: However, it is a unique perspective, well reported, and deserves an audience and feedback. I hope it will be read and responded to.

AUTHOR: Thank you

REVIEWER: It is quite well written with a few weak spots:“3. Spiritual Care…” line 49 --- poor syntax

AUTHOR: The sentence has been rewritten:

Pg 2 line 49: ‘In the Dutch Guideline Spiritual Care in Palliative Care, a model is shown [fig 1] that represents the relationship between the four dimensions of palliative care

REVIEWER Line 185 “…needs as well as possible.”

AUTHOR: Thank you, Pg 5 line185: the word ‘good’ has been replaced by ‘well’

Reviewer 2 Report

This paper, from a physician, shows how important it has become to move beyond the physical symptoms, even as palliative care is primarily focused on pain relief. The author reviews attempts to divide "pain" into 4 aspects, which I believe ultimately derives from Dr. Cecily Saunders and Hospice. But then this author acknowledges that such division, and various charts, do not do justice to the intertwined nature of the patient's own experiences and narrative. What this author as a "spiritual generalist" opts for a a holistic openness to whatever the patient considers relevant. An alternative would have been to "re-specialize," sending the patient to a chaplain or therapist for "spiritual care" - something this author avoids.

One reality behind this whole discussion is that the concept of "spiritual" and "spirituality" is of necessity very vague and murky. Frankl, whose work is cited, wrote as a psychologist, but the range and scope of psychology has shifted, and "spiritual" now fills in the niche once occupied by humanistic psychologies. The author is correct that spending time trying to separate spiritual from psycho-social dimensions of pain or care, is off the mark. There is no way to separate firmly, just as trying to separate spiritual from religious is a murky and futile task. "Spiritual" may be useful here precisely because it is not precise, but murky, covering a variety of bases.

One issue to raise here is about the logistics of doing this kind of holistic care in an understaffed, overworked hospital setting, where there may be patients from a wide range of cultures and ethnicities. Even the most open and sensitive palliative care specialist may find it impossible to do justice to this ideal of holistic care under these circumstances, very common in large city hospitals in the USA.
